# Electrochemical-Based Biosensor Platforms in Lab-Chip Models for Point-of-Need Toxicant Analysis

Mohana Marimuthu [1,2,*], Vinoth Krishnan [3,4], Shailendra Devi Sudhakaran [3], Sevakumaran Vigneswari [5], Shanmugam Senthilkumar [3,4] and Murugan Veerapandian [3,4,*]

1. Centre for Research, Dhanalakshmi Srinivasan University, Samayapuram 621112, Tamil Nadu, India
2. Department of Biomedical Engineering, Dhanalakshmi Srinivasan University, Samayapuram 621112, Tamil Nadu, India
3. Electrodics and Electrocatalysis Division, CSIR-Central Electrochemical Research Institute (CECRI), Karaikudi 630003, Tamil Nadu, India; vinoth.cecri21a@acsir.res.in (V.K.)
4. Academy of Scientific & Innovative Research (AcSIR), Ghaziabad 201002, Uttar Pradesh, India
5. Institute of Climate Adaptation and Marine Biotechnology (ICAMB), Kuala Nerus 21030, Terengganu, Malaysia
* Correspondence: mohana.3m@gmail.com (M.M.); vmurugan@cecri.res.in (M.V.)

**Abstract:** The global hazardous waste management market is expected to reach USD 987.51 million by 2027 at a CAGR of 14.48%. The early detection of corrosive, flammable, and infectious toxicants from natural sources or manmade contaminants from different environments is crucial to ensure the safety and security of the global living system. Even though the emergence of advanced science and technology continuously offers a more comfortable lifestyle, there are two sides of the coin in terms of opportunities and challenges, demanding solutions for greener applications and waste-to-wealth strategies. A modern analytical technique based on an electrochemical approach and microfluidics is one such emerging advanced solution for the early and effective detection of toxicants. This review attempts to highlight the different studies performed in the field of toxicant analysis, especially the fusion of electrochemistry and lab-chip model systems, promising for point-of-need analysis. The contents of this report are organised by classifying the types of toxicants and trends in electrochemical-integrated lab-chip assays that test for heavy-metal ions, food-borne pathogens, pesticides, physiological reactive oxygen/nitrogen species, and microbial metabolites. Future demands in toxicant analysis and possible suggestions in the field of microanalysis-mediated electrochemical (bio)sensing are summarised.

**Keywords:** hazardous material; toxic detection; μTAS; lab-on-chip; early detection; electrochemistry

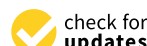



## 1. Introduction

An electrochemical biosensor is an integrated system involving a redox-species-mediated reaction with a biorecognition element or bioreceptor, transforming signals based on a principle like an enzymatic biochemical process, antigen–antibody interaction, nucleic acid hybridisation, or the crosslinking of biomimetic macromolecules [1]. An electrochemical biosensor system can be designed to provide analytical information in different forms, such as yes/no responses, and in a quantitative or semi-quantitative manner. The global market potential of electrochemical biosensors is progressing to a large extent, from USD 16.9 billion in 2023 to USD 28.3 billion by 2032, with a compound annual growth rate of 6.65% for the forecast period (2023–2032) [2]. This proclaimed market growth in different industries is certainly interlinked with rising disease threats and environmental pollution. The World Health Organization encourages research on point-of-care diagnostics with the goal of meeting the ASSURED criteria, which stands for Affordable, Sensitive, Specific, User-friendly, Rapid and Robust, Equipment-free, and Deliverable to end users [3]. Common bottlenecks in conventional laboratory analyses are the practical difficulties resulting

from sample collection and analysis at different locations, apparently influencing the diagnosis of the first clinical symptoms and the selected therapeutic regimen. Instances of false negatives and delayed hazard mitigation alerts in the case of environmental contaminants also drive research on alternative analytical techniques.

Although urbanisation and industrial development have improved the quality of human life, the consequences arising from environmental toxicants are a crucial issue that needs immediate attention. There are significant recent reviews covering the widespread utility of an electrochemical sensor strategy for water pollutant analysis: for instance, the electrochemical-based sensing of water toxicants such as pharmaceuticals, endocrine-disrupting compounds, personal care cosmetics, fire retardants, solvents, pesticides and heavy metals, antibiotics, surfactants, and preservatives [4]. Due to the growing number of advancements in pharmaceuticals, agro-livestock management, and the food processing/packaging industries, the role of emerging fine chemicals or specialty compounds is inevitable. Recent studies have documented that emerging water contaminants have highly complex effects, even at residual concentrations ranging from ng/L to µg/L.

Similar to chemical contaminants, infectious toxicants are huge health hazards. Pre-pandemic records suggest that 6 out of 10 instances of global mortality are directly or indirectly related to infectious diseases, and they are ranked among the top five leading contributors to the global death ratio [5]. Airborne submicroparticle matter from industrial flue gas or vehicle exhausts often tends to weaken the immune system, apparently affecting the respiratory system via infections/inflammation like tuberculosis, chronic obstructive pulmonary disease, neonatal infection, etc. [6,7]. At the cytophysiological level, recurring exposure to toxic pollutants may damage or kill epithelial cells, ultimately causing the collapse of the integrity of the alveolar–capillary barrier and leading to chronic lung inflammation so that the inherent defence function of the lungs against infectious pathogens is not feasible.

Ensuring the safety of the soil, water, and air is an integral right of the global population. To circumvent the challenges in conventional toxic contaminant analysis, several alternative techniques have been explored using modern sensor strategies. For instance, an electrochemical method possesses advantages due to its portable device development and amplified signal transduction, even at ultralow sample concentrations. Depending on the experimental conditions and complexity of the test samples, electrochemical techniques can be optimised to achieve a rapid analytical response time. The fundamental methods of analysis for exploring the basic electrochemical characteristics of a redox-active system intended for biosensor studies are voltammetry and electrochemical impedance spectroscopy. In any electrochemical analysis, the magnitude of the detection parameters can be influenced by the method for probing the Faradaic current response. In this respect, square-wave and differential pulse voltammetric techniques are comparatively more sensitive than cyclic voltammetry/linear sweep voltammetry. The impact of the electrochemical sensor on analytical chemistry has opened up new horizons in environmental safety and security.

Amongst the different innovative technologies, microelectromechanical system (MEMS)-based microfluidics has emerged as a valuable tool to provide solutions for analytical science research problems. Externally controllable micropumps and valves integrated with electronic, electrochemical, and piezoelectric/magnetic sensors and actuators are emerging innovations in microfluidic-based analytical applications [8–11]. The amalgamation of electrochemical techniques into a lab-on-chip (LOC) environment makes the detection platform more suitable for handling a low volume of reagents, a precise target–bioreceptor interaction, and a rapid analytical response. Articles detailing the basic definitions and principles of electrochemistry, biosensors, microfluidics, and microfluidic electrochemical biosensors for point-of-care diagnostics are available [12–14]. A review of different point-of-care diagnostics for infectious diseases and their different governing principles, *viz.*, surface plasmon resonance, chemiluminescence, colorimetric, fluorescence, surface-enhanced Raman, and magnetic biosensors, in addition to electrochemical bioassays with device applications, exists in [15]. Efforts were also devoted to reviewing the challenges

in automated sample solution preparation integrated with nucleic acid amplification and rapid/precise home-care diagnostics [16]. In this review, we analyse the different types of toxicants, the conventional methods of analysis deployed in toxicant investigations, and recent trends in electrochemical biosensor platforms integrated with lab-chip models demonstrated for infectious-disease-causing toxicants. We hope that this report will present a framework to address the need for the essential integration of electrochemical biosensors for convenient sample preparation with isolation/amplification-free signal transduction, which is promising for advanced point-of-need analysis.

## 2. Classification of Toxicants

An agent that causes significant adverse effects or seriously damages the regular physiological function of biological systems, leading to lethality, is generally regarded as a toxicant [1]. It can be classified by its source of origin, physicochemical state or effect, the target site of action (enzyme or substrate), biological behaviour, and nature of use. Manmade products from industries such as food preservatives, particulate pollutants from automobiles, hazardous biowaste, radioactive substances, antibiotics, fertilisers, pesticides, and dyes are also toxic to individuals and the environment. In this section, different types of plant- and marine-derived toxins are described in terms of their source, lethal concentration, and mode of conventional analysis (Table 1). It is evident that conventional spectroscopic and chromatographic techniques are largely utilised for the detection and quantification of different toxins. Considering the source of their existence (agri-livestock products) and the severity of their exposure, it is absolutely essential that anyone can test for their presence on the spot. In this respect, the modern biosensor approach, with or without sample pretreatment, largely simplifies analytical procedures. In the following section, the recent trends in electrochemical methods for detecting important toxicants are discussed. Though electrochemical methods offer a rapid analytical response and portable device/gadget fabrication, there are still challenges, including sample handling at the electroactive surface, better loading of bioreceptors, the inability to sort the target from interferants, multiplexed detection, and simplified data readouts. With a micro total analysis strategy, i.e., a microfluidic environment, the flow of analyte samples on the bioactivated substrate can be specifically improved for selective detection. To highlight such fusion technology, selected research studies demonstrating the feasibility of lab-chip models in toxicant detection based on electrochemistry principles are reviewed.

**Table 1.** Classification of toxicants from different plant and marine sources.

| Toxicant | Toxic Dose/ Concentration | Source | Monitoring Method | References |
|---|---|---|---|---|
| Cyanogenic glucoside | 15 mg/kg | Cassava | NMR | [17] |
| Glycoalkaloids | 20 mg/100 g | Potato | LC-MS/MS | [18] |
| Phytohemagglutinin | 20,000 to 70,000 hau | Red kidney beans | LC-MS/MS | [19] |
| Ricin | 1 to 20 mg/kg | Castor beans | Immunocapture and MALDI-TOF/MS | [20] |
| Glycoside amygdalin | 20 mg/100 g | Almond | $^1$H-NMR and $^{13}$C-NMR | [21] |
| Latrunculin | Human inhalation toxic concentration low (TCLO) 20 m: 2500 mg/m$^3$ | *Negombata magnifica* (marine sponges) | HPLC | [22] |
| Tetrodotoxin | 334 µg/kg | Puffer fish | HPLC; LC-MS; GC-MS | [23] |
| Histamine | 100 mg/kg | Raw/Chilled/ Frozen Finfish | HPLC | [22] |

**Table 1.** *Cont.*

| Toxicant | Toxic Dose/ Concentration | Source | Monitoring Method | References |
|---|---|---|---|---|
| Paralytic shellfish toxins | 80 μg/100 g | Bivalve molluscs | UPLC-MS/MS | [24] |
| Azaspiracid | 160 μg/Kg | Bivalve molluscs | LC-MS/MS | [25] |
| Brevetoxins | 520 μg/Kg | *Karenia brevis* | LC-HRMS | [26] |
| Yessotoxins | 25 μg YTX equivalents/Kg body weight | *Lingulodinium polyedrum* | HPLC-FLD | [27] |
| Saxitoxins, Neosaxitoxin, Decarbamoyl saxitoxin, and Gonyautoxin 1 | Saxitoxin contents in cyanobacterial biomass 4470 μg/g; <10 μg/L in drinking water | Freshwater fish, molluscs, and crayfish; drinking water; recreational activities in lakes and rivers | HPLC/LC-MS; lateral flow immunoassay; ELISA; electrochemical immunoassay; Radio Immunoassay | [28–30] |

Note: hau = haemagglutinating unit; LC-MS = Liquid Chromatography–Mass Spectrometry; NMR = Nuclear Magnetic Resonance; MALDI-TOF/MS = Matrix-Assisted Laser Desorption Ionisation–Time-of-Flight Mass Spectrometry; $^1$H = Proton; $^{13}$C = Carbon; HPLC = High-Performance Liquid Chromatography; GC-MS = Gas Chromatography–Mass Spectrometry; UPLC-MS/MS = Ultra-Performance Liquid Chromatography–Tandem Mass Spectrometry; LC-HRMS = Liquid Chromatography–High-Resolution Mass Spectrometry; HPLC-FLD = Liquid Chromatography with Fluorescence Detection; ELISA = Enzyme-Linked Immunosorbent Assay.

## 3. Trends in Electrochemical Biosensors with Lab-Chips

Owing to their low analyte-sample-handling requirements and their superior sensitivity and selectivity, lab-chip-integrated electrochemical biosensor platforms are attractive for rapid analytical applications [31,32]. Electrochemical biosensors can be embedded in microfluidic channels using advanced microfabrication technologies, and the miniaturisation of the potentiostat with a smartphone readout enables the laboratory-based analysis of samples in the field [12,33,34]. Figure 1 provides a general overview of an electrochemical biosensor configuration on a microfluidic platform. Customised bioreceptors like antibodies, enzymes, biosimilars (molecularly imprinted polymers), and nucleic acids are the range of active biorecognition elements widely studied for electrochemical biosensor design and application. The factors involved in the fusion technology include (i) the design of the electrode layout, microfluidic channel, and bioreceptor immobilisation, (ii) the fabrication of compatible substrates (e.g., PDMS) and the selection of techniques for the creation of the microfluidic path (e.g., photolithography) and electrode deposition (e.g., chronoamperometry), (iii) the integration of fluidic and electrode layers to create a functional lab-chip with biomolecular hybridisation, followed by fluidic and electrochemical connections, (iv) the testing and validation of the lab-chip using standard samples to evaluate its sensitivity, specificity, and detection range [35,36]. Whiteside and Liang's group developed a paper-based electrochemical biosensor with a microfluidic layer for monitoring environmental pollutants and organic pesticides [37,38]. This strategy was also demonstrated with conventional glassy-carbon-based electrodes in the flow-assisted mode to detect marine toxicants. Figure 2 shows the miniaturised potentiometric biosensor platform developed for testing saxitoxin (STX) using anti-STX immobilised on a lipid layer on a graphene nanosheet electrode based on the immunocomplex principle. The as-developed biosensor exhibits better selectivity and sensitivity, enabling a detection limit of 1 nM, with a rapid response of 5–20 min.

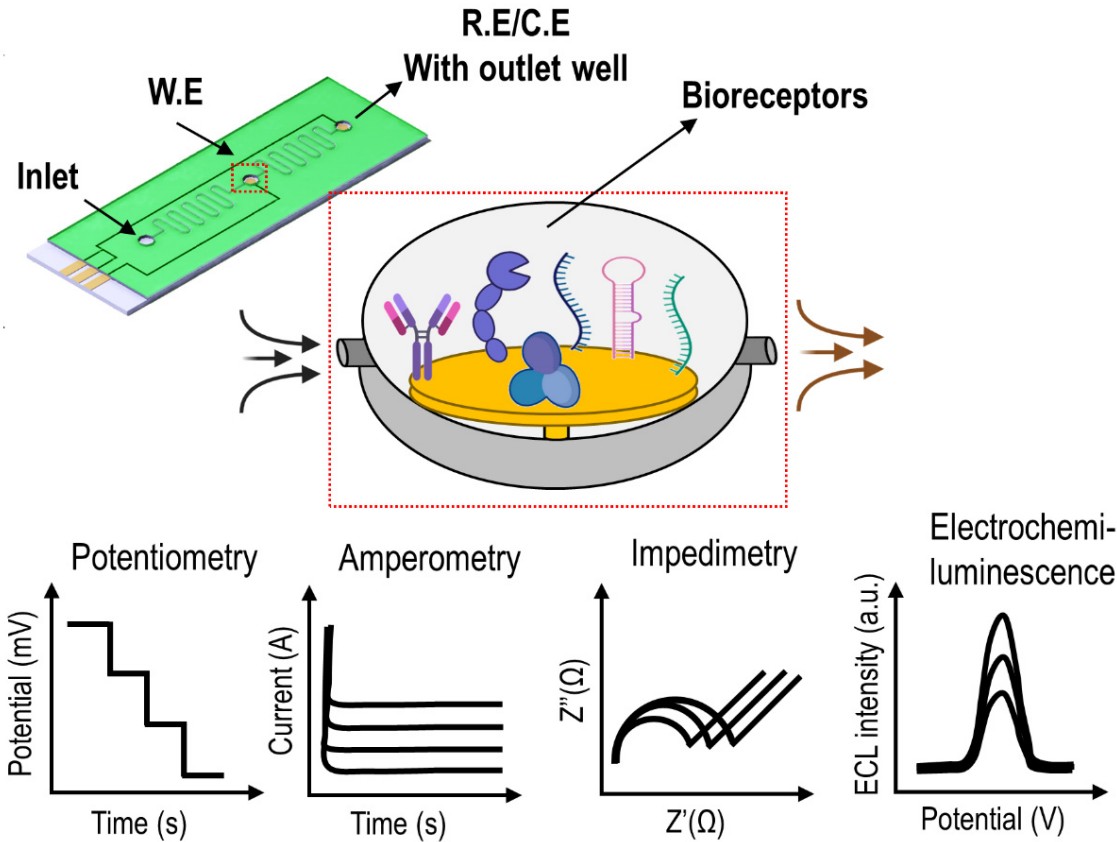

**Figure 1.** Illustration of electrochemical biosensor configuration on a microfluidic platform and various detection techniques.

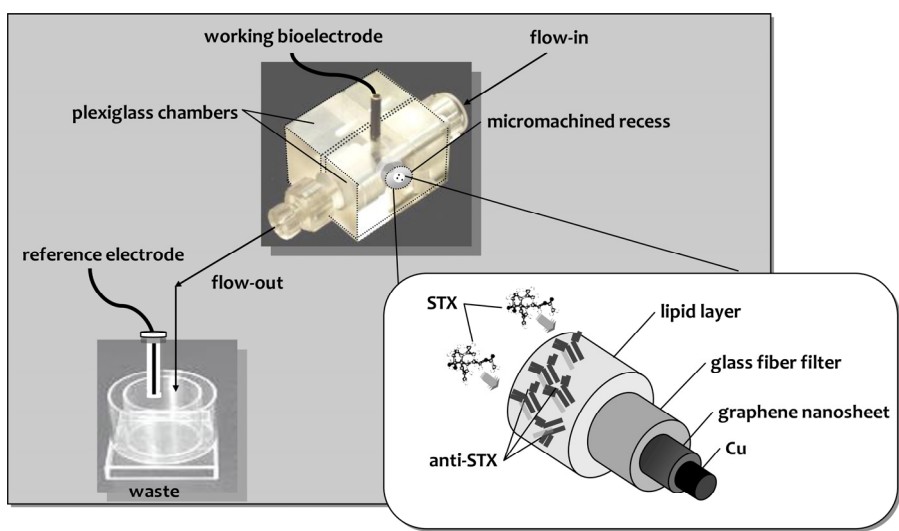

**Figure 2.** Schematic configuration of miniaturised potentiometric saxitoxin biosensor (adapted with permission from [30] copyright 2023, John Wiley and Sons).

## 3.1. Heavy-Metal Toxicant Detection

The electrochemical sensing of heavy-metal ions in wastewater is a crucial and environmentally significant application to prevent toxic effects. Heavy metals, such as lead, cadmium, mercury, and arsenic, are toxic pollutants commonly found in industrial wastewater, posing serious threats to both human health and the ecosystem [39,40]. In this regard, the World Health Organization (WHO) and the U.S. Environmental Protection Agency (US-EPA) have provided guidelines on the maximum contaminant levels of metals in

drinking water. The heavy-metal concentrations in water should not exceed 2, 5, 15, and 100 µg/L for Cd, Hg, Pb, and Cr, respectively [41,42]. Traditional methods like atomic absorption spectroscopy [43], inductively coupled plasma–atomic emission spectroscopy [44], and inductively coupled plasma–mass spectroscopy [45] for heavy-metal ion detection in wastewater involve complex and time-consuming processes, often requiring sophisticated equipment and trained personnel. In contrast, electrochemical sensors offer a rapid, cost-effective, and highly sensitive approach for the real-time monitoring of heavy-metal contaminants [39]. Table 2 shows the different analytical reports demonstrating heavy-metal pollutant monitoring.

Different biological recognition elements, such as enzymes, antibodies, or DNA sequences, were studied for the specific detection of particular heavy-metal ions [46]. These elements are immobilised onto the surface of an electrochemical transducer, typically a working electrode made of conductive materials like gold or carbon. For instance, an enzyme-based biosensor was shown to have the potential to detect heavy metals such as $Pb^{2+}$, $Cu^{2+}$, $Cd^{2+}$, Cr (VI), and $Hg^{2+}$ [47–49]. This biosensor exhibits high specificity and repeatability and a low detection limit; however, maintaining the stability of the enzyme could be challenging. Li et al. developed a microbial biosensor comprising *E. coli* immobilised with benzoquinone within a gelatin/silica hydrogel on the glassy carbon surface (abbreviated BGSH). The *E. coli*-modified electrode matrix mediates the metabolic reaction with the electrolyte (respiratory substrate or nutrient) and augments the reduction/re-oxidation of the quinone moiety within the BGSH redox-active hydrogel. During the addition of toxic metal ion pollutants, the microbial population on the sensor surface undergoes detrimental effects, ultimately influencing the redox behaviour of the quinone groups on the electrode. Through this co-immobilisation strategy, the biotoxicity of $Hg^{2+}$, $Cu^{2+}$, and $Cd^{2+}$ in water was determined to be 21.2, 44, and 79 µg $mL^{-1}$, respectively [50]. In another study, the cytotoxicity of heavy-metal ions (Cr, Cd, Cu, Pb, and Zn) was tested in human cervical carcinoma (HeLa) cells using electrochemically reduced graphene oxide [51]. The catabolic intermediates of HeLa cells, i.e., intracellular guanine and xanthine, exerting irreversible anodic peak potentials at +0.681 and +0.689 V, are probed to correlate the viability of cells before and after exposure to the aforesaid heavy-metal ions. An electrochemical-based DNA biosensor was also deployed for the detection of heavy-metal ions by using single-stranded DNA as a probe to modify $Fe_3O_4$@AuNPs based on the metal-mediated base-pairing principle [52]. Miao et al. reported the stable duplex formation of DNA bases using mismatched base pairs (C-C) and (T-T), which form a stable duplex with other DNA probes (labelled with methylene blue and ferrocene) through coordination chemistry in the presence of specific heavy metals, T-$Hg^{2+}$-T and C-$Ag^+$-C. The hybridisation of redox-labelled DNA probes was electrochemically correlated for the nanomolar-level detection of $Ag^+$ and $Hg^{2+}$ by using square-wave voltammetry. Probe DNA activation on the sensor surface is based on thiol-Au chemistry. The utilisation of a magnetic glassy carbon electrode substrate enabled the firm immobilisation of the $Fe_3O_4$@Au sensor element without complex modification.

**Table 2.** Various electrochemical-integrated lab-chip assays of heavy-metal toxicants.

| Target Analyte | Classification Analyte | Method of Detection | Recognition Element | Detection Limit | Toxic Dose (mg/L) | Reference |
|---|---|---|---|---|---|---|
| $Pb^{2+}$, $Ni^{2+}$, $Cd^{2+}$ | Inorganic | Amperometry | Horseradish peroxidase | 8.0, 3.0, 1.0 nM | 0.010 0.07 0.003 | [53] |
| $Hg^{2+}$ | Inorganic | Amperometry | Catalase | $1.8 \times 10^{-11}$ M | 0.001 | [54] |
| $Hg^{2+}$, $Cd^{2+}$, $Pb^{2+}$, Cr(VI) | Inorganic | Amperometry | Glucose oxidase | 2.3 nM, 1.75 nM, 2.70 nM, 2.44 nM | 0.001 0.003 0.01 0.05 | [55] |

**Table 2.** *Cont.*

| Target Analyte | Classification Analyte | Method of Detection | Recognition Element | Detection Limit | Toxic Dose (mg/L) | Reference |
|---|---|---|---|---|---|---|
| Cr(VI), Cr (III) | Inorganic | Amperometry | Glucose oxidase/ horseradish peroxidase | 0.20 nM, 0.01 μM | 0.05 | [48] |
| $Zn^{2+}$ | Inorganic | Cyclic voltammetry | *E. coli* BL21 | 20 μM | 3 | [56] |
| $Cu^{2+}$ | Inorganic | Picoammeter– voltage (I–V) | Biotinylated substrate strand (S strand) and catalyse strand (C strand) were assembled with cDNAzyme | 100 pM | 2 | [57] |
| $Hg^{2+}$ | Inorganic | Impedance | Manganese porphyrin-decorated DNA network | 1.47 pM | 0.001 | [58] |
| Chlorpyrifos and $Pb^{2+}$ | Inorganic | Differential pulse voltammetry | DRAB | 0.178 nM and 0.034 nM | 0.05 0.01 | [59] |

Note: $Pb^{2+}$ = lead; $Ni^{2+}$ = nickel; $Cd^{2+}$ = cadmium; $Hg^{2+}$ = mercury; Cr = chromium; $Zn^{2+}$ = zinc; $Cu^{2+}$ = copper; nM = nanomolar; μM = micromolar; pM = picomolar; *E. coli* = *Escherichia coli*; DNA = deoxyribonucleic acid; DRAB = designed Dual-Recognition Aptazyme Beacon.

### 3.2. Pesticide and Food Toxicant Analysis

Pesticides are widely used for healthy crop cultivation to fulfil global needs; however, the increasing usage of toxicants, including herbicides, fungicides, and insecticides, will affect the environment as well as the cultivator [60]. Residual pesticides in agricultural products and soil will eventually spoil the food chain and ground or surface water. In addition to the different types of chromatographic techniques, there are several electrochemical biosensor platforms with lab-chips that have been reported for pesticide and food analysis/monitoring [61,62]. In the case of enzyme-based biosensor studies, Arduini et al. developed a paper-based 3D electrochemical device for the detection of several pesticides in river water. The multiplex analysis comprised a carbon black/Prussian blue nanocomposite substrate followed by an individual enzyme loading for each pesticide's detection, i.e., paraoxon to inhibit butyrylcholinesterase, 2,4-dichlorophenoxyacetic acid for alkaline phosphatase, and atrazine towards tyrosinase. The chronoamperometry technique was employed to monitor the enzymatic inhibition, enabling a limit of detection of 2 ppb of pesticides in water sample analysis [63]. In another study, acetylcholinesterase was entrapped by glutaraldehyde immobilised on a single-walled carbon nanotube-enclosed bovine-serum-albumin-modified electrode for the selective detection of methyl parathion (organophosphorus pesticide) using the cyclic voltammetry technique. This biosensor exhibits a wide linear range from $1 \times 10^{-10}$ M to $5 \times 10^{-6}$ M, with a limit of detection (LOD) of $3.75 \times 10^{-11}$ [64]. Similarly, other enzymes, such as organophosphorus hydrolase, phosphotriesterase, butyrylcholinesterase, and choline oxidase, were also demonstrated for the electrochemical detection of pesticides [65,66].

Another notable study on an electrochemical lab-chip biosensor platform was based on whole-cell-targeted pesticide detection. Microorganism-immobilised biosensors are advantageous for their rapid analysis of toxicants because of their constrained bacterial growth upon exposure to pesticides. Pabbi et al. developed an algal biosensor on silica-coated ZnO quantum dots for acephate pesticide detection. This biosensor works on the basis of the production of *p*-nitrophenol by the dephosphorylation of *p*-nitrophenylphosphate using *Chlorella* sp. algal cells, and it exhibits a linear range of $10^{-11}$ M to $10^{-3}$ M with a limit of detection of $1.0 \times 10^{-12}$ M [67]. Tucci et al. reported an amperometric biosensor for atrazine and diuron detection through *Anabaena variabilis* inhibition. This biosensor

was constructed using bacterial-cell-entrapped alginate on a carbon-felt electrode with *p-benzoquinone* as the redox probe. The working principle is based on the inhibition of the photocurrent generated by the microorganism upon herbicide interaction, and the obtained LOD was 0.07 μM [68].

Similarly, many studies have been carried out on electrochemical biosensors with lab-chip models for detecting microorganisms or their metabolites (toxicants) in food samples. The major reports focused on monoclonal antibodies as recognition elements for pathogen-detection voltammetric and impedimetric techniques (Table 3).

**Table 3.** Electrochemical-biosensor-based LOC assays of pesticide and food contaminants.

| Target Analyte | Classification of Analyte | Method of Detection | Recognition Element | Detection Limit | Toxic Dose (mg/L) | Reference |
|---|---|---|---|---|---|---|
| Carbendazim, Chlorpyrifos, DDT, Dinocap, Ethion | Organic | Chronoamperometry | Glutathione-S-transferase | 20 ppb 60 ppb 40 ppb 50 ppb 100 ppb | 50 mg/L 0.05 mg/L 0.001 mg/kg 120–140 mg/kg 2 mg/kg | [69] |
| Dichlorvos | Organic | Amperometry | Choline oxidase enzyme | 1.6 nM | 1 mg/kg | [66] |
| Paraxon | Organic | Amperometry | Phospho triesterase | 3 nM | 0.5 mg/kg | [70] |
| Glyphosate 2,4-dichlorophenoxyacetic acid (2,4-D) | Organic | Differential pulse voltammetry | DNA | ** | 40–50 mg/kg | [71] |
| Carbofuran | Organic | Differential pulse voltammetry | Molecularly imprinted film (MIP) and a DNA aptamer as dual-recognition element | $6.7 \times 10^{-11}$ mol·L$^{-1}$ | 0.1 mg/kg | [72] |
| **Food sample analysis** | | | | | | |
| *Salmonella typhimurium* | Biological | Differential pulse voltammetry | Monoclonal anti-salmonella (Ab$_1$) and polyclonal anti-salmonella (Ab$_2$) antibodies | 7.7 cells mL$^{-1}$ | ** | [73] |
| Mycotoxins: Fumonisin B1 (FB1) and Deoxynivalenol (DON) | Biological | Differential pulse voltammetry | Anti-FB$_1$ and anti-DON antibodies | 97 pg/mL and 35 pg/mL | ** | [74] |
| *Salmonella typhimurium* | Biological | Electrochemical impedance spectroscopy | Anti-*S. typhimurium* | 1.56 CFU/mL | ** | [75] |
| *Listeria monocytogenes* | Biological | Electrochemical impedance spectroscopy | Antibodies specific for *L. monocytogenes* | 5.5 cfu/mL | ** | [76] |
| Dimethoate | Biological | Amperometry | Acetylcholinesterase | 4.1 nM | ** | [77] |
| *Salmonella typhimurium* | Biological | Amperometry | Polyclonal antibody specific to salmonella | 10 CFU mL$^{-1}$ | ** | [78] |
| Norovirus | Biological | Differential pulse voltammetry | Monoclonal antibody | $10^4$ genomic copies/mL | ** | [79] |

Note: DDT = Dichloro-Diphenyl-Trichloroethane; CFU = Colony-Forming Unit; pg = Picogram; mL = Millilitre; ** = not available.

### 3.3. Reactive Oxygen Species as Molecular Toxicants

Apart from environmental bioprocess-mediated toxins and manmade pollutants, physiological systems can also create toxic effects resulting from free radicals (reactive oxygen species, ROS, and reactive nitrogen species, RNS) due to metabolic collapse or exposure to toxins apparently damaging cellular function and leading to chronic conditions, like cardiovascular diseases, inflammatory diseases, and cancer [80]. This section highlights

the demonstrated electrochemical and electrochemiluminescence methods with microfluidic platform integration for detecting ROS/RNS from different substrates. For instance, Gomez et al. fabricated an electrode system with microfluidic interconnections for monitoring cellular behaviour. This multiplexed system detects the analytes, namely, glucose, hydrogen peroxide ($H_2O_2$), conductivity, and oxidation–reduction potential (ORP), in an independent chamber. Experiments performed using individual techniques exhibit better sensitivity for each analyte, *viz.*, $1.8 \pm 0.2$ µA mM$^{-1}$ for $H_2O_2$ using the chronoamperometry technique, $0.14 \pm 0.01$ mV $\Omega^{-1}$ cm$^{-1}$ for conductivity based on the bipolar method, and $0.0060 \pm 0.0003$ µA mM$^{-1}$ for glucose monitoring using the cyclic voltammetry technique [81]. In another study, three-dimensional silver foam (Ag-wire foam) was developed to detect $H_2O_2$ in three different cancer cells, namely, human leukaemia K562, human cervical cancer (Hela), and MCF-7 cells. Among them, the Ag-based lab-chip model exhibits a low detection limit of 15 nM in human leukaemia K562 cancer cells [82]. The microfluidic platform augmented the upstream construction of channels for the culture of cells and parallel detection chambers for multiple ROS/RNS probing, *viz.*, $H_2O_2$, ONOO$^-$, NO$^\bullet$, and NO$_2$. Using the amperometric technique, calcium-ionophore-triggered RAM 264.7 macrophage secretions against oxidative stress were assessed in an integrated microfluidic device [83].

It has been established that aerobic metabolism spontaneously produces ROS from 1% to 3% of the oxygen it uses. One of these species, hydrogen peroxide, is the most persistent ROS and may permeate to practically any cellular compartment, in addition to being stored in significant amounts in the cell, making it a significant hydroxyl radical source. The hydroxyl radical is a potent and hazardous oxidant molecule with high reactivity and a brief lifetime, and it typically interacts with the first molecule it encounters right where it forms [84]. The hydroxyl radical can have a wide range of harmful biological effects, including oxidising or hydroxylating proteins or DNA bases, peroxiding cell membrane bilipids, and more [85,86]. A significant human toxicity pathway known as genotoxicity involves damage to DNA caused by drug metabolites and pollutants. Through a process known as bioactivation, medications and environmental pollutants can be metabolically converted by cytochrome P450s (cyt P450) and other enzymes in humans to metabolites that are chemically reactive. Reactive oxygen species (ROS) are created when some redox-active metabolites, metal ions, and NADPH combine to oxidise DNA and create covalently bound adducts with DNA bases [87,88]. Recently, Bist et al. [89] reported a novel two-channel microfluidic ECL array with 30 microwells that can identify both oxidised and metabolite-adducted DNA (Figure 3).

Four aryl amines, whose metabolites cause both DNA oxidation and nucleobase adduction, were used to test these novel arrays. The arrays showed the ability to detect both types of DNA damage and assess the impact of the bioactivation of these aryl amines by various human cytochrome P450s. Toxicology bioassays are important tools to complement in vitro screening assays that reveal potentially genotoxic chemistry pathways to forecast drug and pollutant toxicity. Microwell arrays containing reactant solutions in the presence of DNA/enzyme films at an applied potential generate metabolites. With the subsequent NADPH catalytic reaction, ROS are generated within the array environment via the activation of cyt P450s. As-generated reactive metabolites then form DNA adducts, which are detected through the RuPVP-based ECL system. Oxidative DNA damage from a $Cu^{2+}$-metabolite-mediated redox pathway was detected by ECL using [Os(bpy)$^2$(phenbenz-COOH)]$^{2+}$. Calibration for the oxidation product 8-oxodG was validated using LC/MS/MS [89]. With the selective modification of sensor elements on a single electrode or an array of electrodes, one can personalise the specific detection of molecular toxicants such as ROS/RNS, even at the cellular level.

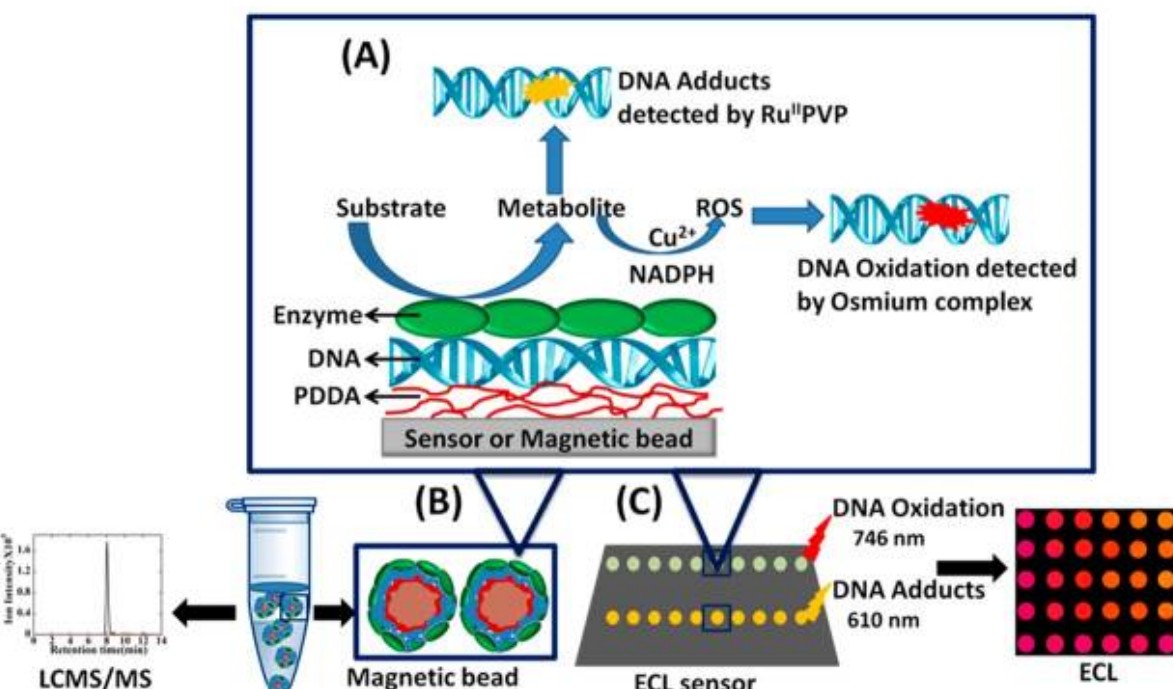

**Figure 3.** Array strategy for screening genotoxic pathways using (**A**) DNA/enzyme films, (**B**) Biocolloid Reactors, and (**C**) ECL arrays (adapted with permission from [89] copyright 2023, ACS).

*3.4. Microbial Metabolites and Antibiotic Resistance Test*

Metabolites play a crucial role in various biological processes, including energy production, cell growth, and interactions with other organisms. Some microbial metabolites also possess pharmaceutical potential, serving as the basis for the development of antibiotics, antifungals, and other therapeutic agents. Understanding microbial metabolites is essential for diagnosing and treating infections, as they can serve as biomarkers for the presence of specific pathogens. Parallelly, microbial metabolites can also cause lethal effects on human health. Fungal metabolites are often known to have broad-spectrum antimicrobial applications. Thus, testing the therapeutic efficacy of antibiotics is vital in modern medicine for combating bacterial infections. The emergence of antibiotic-resistant bacteria has become a global health crisis. To combat this threat effectively, it is essential to determine the susceptibility of bacterial pathogens to antibiotics rapidly and accurately. Traditional methods of antibiotic susceptibility testing are often time-consuming and labour-intensive, delaying the initiation of appropriate treatment and contributing to the spread of antibiotic resistance. In this context of microbial metabolites and antibiotic susceptibility testing, an integrated LOC model can automate the cultivation of microbial cultures, monitor their growth, and simultaneously analyse the production of specific metabolites. It can also perform high-throughput antibiotic susceptibility testing, enabling healthcare providers to quickly identify the most effective antibiotics for specific infections. Assay models have been developed based on electrochemical-biosensor-coupled microfluidic patterns for monitoring microbial toxicants. Mycobacterium within the *Actinobacteria* genus is regarded as one of the most common infectious-disease-causing pathogens in mammals, causing diseases such as tuberculosis and leprosy. Due to its adaptable yet complex cell wall characteristics, Mycobacterium has developed multiple strategies and resistance mechanisms to infect the human population. The detection of the lysis and fragmentation of the cell or an associated component following the incubation of the pathogen with a controlled substrate is a common principle often explored in lab-chip technology. A combination of an electrode array consisting of a gold transducer platform with a specific nucleic acid probe sequence against the 16SrRNA region of *Mycobacterium smegmatis* was tested using DPV and SWV [90].

Aflatoxin $B_1$ (AFB$_1$) is one of the highly toxic difuranocoumarin derivatives, often produced by the fungi *Aspergillus flavus* and *A. parasiticus,* existing in a wide range of agricultural foods and animal feedstuffs due to storage temperatures and humid environments accelerating mould growth. Among the four major aflatoxins, i.e., B1, B2, G1, and G2, the International Agency for Research on Cancer (IARC) has classified AFB1 as a group 1 human carcinogen, while the other toxins are classified as group 2 human carcinogens. These toxins are also known for teratogenicity and mutagenicity. The electrochemical-immunoassay-principle-based detection of AFB$_1$ in barley [91] and milk [92] was first reported on single screen-printed electrodes. With the advent of integration technology, efforts were further escalated to realize multichannel readouts based on the indirect ELISA format to detect competitive binding between the target AFB$_1$-BSA conjugate and the free AfB$_1$ available on the electrode surface for the binding sites of the anti-aflatoxin B1 antibody. The amount of anti-aflatoxin B1 antibody bound to the immobilised AFB$_1$-BSA was transduced using a secondary antibody labelled with alkaline phosphatase based on intermittent pulse amperometry. With a 96-well screen-printed microplate platform, researchers were able to showcase a multichannel assay for AfB$_1$ in corn samples with a detection limit of 30 pg/mL in the analyte working range between 0.05 and 2 ng/mL. The electrochemical immunoassay was based on the working potential to detect 1-naphthol, the oxidised product of the enzyme/substrate alkaline phosphatase/$\alpha$-napthylphosphate, attached to the anti-aflatoxin B1 antibody [93]. Efforts to develop an integrated chip composed of electrodeposited gold structures with the electrocatalytic reporter pair Ru(NH3)$_6^{3+}$ and Fe(CN)$_6^{3-}$ were demonstrated for the detection of pathogenic bacteria and antibiotic resistance markers using peptide nucleic acid as a bioreceptor. The electrode array is termed a solution circuit chip and consists of 100 working electrodes with 30 off-chip contacts. It includes 20 working electrodes and 5 counter/reference electrode pairs with 25 probe wells to augment the manual probe coating. Five separate liquid channels were also integrated to route reference/counter electrodes. The technique adopted to measure the complementary or non-complementary probe hybridisation specific to pathogens (i.e., bacterial lysates DNA)/antibiotic resistance markers associated with, *viz.*, RNA polymerase β mRNA (*rpo*β) or a ribosomal RNA, is differential pulse voltammetry using the aforesaid reporter pairs [94].

*3.5. Strategies to Circumvent the Effects of Matrices*

The selective and specific detection of any analyte is challenging due to common interferents/matrices co-existing in the test samples. There are different approaches explored to address interferent-free electrochemical sensing. For instance, the Cu(II) ion is a common interferent in heavy-metal ion detection due to the formation of intermetallic compounds. The careful addition of complexing agents like ammonia solution to analyte samples can mitigate such matrix effects during As(III) ion detection in water analysis [95]. Similarly, the construction of hierarchically ordered porous structures, such as binary oxides (Ce-Zr), enabled better adsorption of Pb(II), even in the presence of other metal ions, like Hg(II), Cd(II), Cu(II), and Zn(II) [96]. Apart from stripping voltammetric techniques, efforts to devise an electrochemiluminescence (ECL) probe composed of polyluminol with Au NPs (PL-Au) were also demonstrated for the selective detection of toxic Hg$^{2+}$. Interactions with Hg$^{2+}$ ions at the interface of PL-Au accelerate the decomposition of H$_2$O$_2$ (without an additional co-reactant), which linearly increases the ECL signal even in the presence of interferants like Fe$^{2+}$, Co$^{2+}$, Pb$^{2+}$, As$^{2+}$, Cu$^{2+}$, Cd$^{2+}$, Zn$^{2+}$, and Mg$^{2+}$ in test samples, which is promising for pollutant monitoring [97]. Localisation of the passivation layer with selected inert polymer or neutralising complex layers could also largely prevent non-specific interactions at the electrode interface. With a microfluidic-integrated system, analysis is also feasible by performing sequential separation, incubation, and testing, without the matrix effect from complex interferents. In the case of pathogen sensing, the immobilisation of selected bioreceptors, *viz.*, monoclonal antibodies (including nanobodies), aptamers, peptides, and enzymes, can facilitate the specificity towards the target analytes and nullify

the interferents from the test samples. The emerging scope of artificial intelligence (AI) reveals a new horizon in electrochemical biosensors. An embedded AI model (TinyML) on a portable system can discriminate interferent signals from uric acid and ascorbic acid while probing for electrochemically active target neurotransmitters like dopamine and norepinephrine [98]. Some of the reported machine learning algorithms to enhance the performance of electrochemical (bio)sensors include artificial neural networks, dense neural networks, supervised random forest, and feedforward neural networks [99].

## 4. Summary and Future Perspective

This review summarises the emerging role of the electrochemical approach in conjunction with the lab-chip model for toxicant analysis. Focus has been given to different categories of toxicants from plant/marine sources, heavy metals, pesticides, and food contaminants, in addition to molecular toxicants like ROS and microbial metabolites. The complexity of conventional methods of detection, including sample collection, transportation, pretreatment, sophisticated instrumental analysis, and data interpretation by experts, can be circumvented by modern lab-on-a-chip strategies. With the intervention of electrochemical analytical techniques, point-of-need analysis can be escalated into modern toxicant testing by creating a portable device with a rapid analytical response time and sensitivity promising for in situ detection. It is also worth mentioning that the effective selectivity and specificity of electrochemical analysis are always critical. The major influencing factors are the redox behaviour of the sensor element, the pH of the electrolyte and its composition, the analyte sample state, and cross-talk with interferants. The role of the biorecognition element in the electrode–analyte-sample interaction is also an important factor that can alter the detection performance of the biosensor system.

Next-generation electrochemical-integrated LOC assays should focus on the real-time flow of samples, the sorting of analytes, rapid incubation-mediated recognition, and the translation of the signal into a readable output. To achieve such advanced detection platforms for biological toxicants, multidimensional supports are required. Some of the prospective research areas for microbial toxicants include but are not limited to (i) complementary metal oxide semiconducting-system-based sensor systems integrated with microfluidic and electrochemiluminescence, (ii) the direct electrochemical detection of cell-free nucleic acids, (iii) extracellular-vesicle-associated protein/nucleic acid detection, and (iv) nucleic acid isolation/amplification-free detection and the discrimination of variants. The multiplexed detection of microbial populations in sewage, hospital wastewater, and organic/mixtures of particulate toxicants from pharmaceutical/chemical industries' effluent, either qualitative or quantitative, can be extremely useful for global environmental safety.

**Author Contributions:** M.M.: writing—original draft preparation; visualisation; software; and funding acquisition. V.K.: writing—original draft; investigation. S.D.S.; investigation. S.V.: investigation. S.S.: visualisation and resources. M.V.: conceptualisation, supervision, review and editing, and funding acquisition. All authors have read and agreed to the published version of the manuscript.

**Funding:** This work was supported by the DST-SERB-TARE fellowship (Award Ref. No. TAR/2022/000408) and the SERB, ASEAN-India Collaborative research project (CRD/2021/000451).

**Institutional Review Board Statement:** Not applicable.

**Informed Consent Statement:** Not applicable.

**Data Availability Statement:** The data presented in this study are available on request to the corresponding authors.

**Acknowledgments:** M.M. and S.S. acknowledge the DST-SERB-TARE fellowship (Award Ref. No. TAR/2022/000408). V.K., S.V. and M.V. acknowledge the funding support of SERB, ASEAN-India Collaborative research project (CRD/2021/000451). CSIR-CECRI Manuscript Communication Number CECRI/PESVC/Pubs/2023-140.

**Conflicts of Interest:** The authors declare no conflict of interest.

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
