# Peer review of "Electrochemical-Based Biosensor Platforms in Lab-Chip Models for Point-of-Need Toxicant Analysis"

_2673-3293, doi:10.3390/electrochem4040034_

Round 1
Reviewer 1 Report
Comments and Suggestions for Authors
This is a nicely written manuscript from Mohana Marimuthu et al. that presents and summarizes different studies in point-of-need toxicant analysis by electrochemical-based biosensor platform on lab-chip model system. I think this manuscript could be appealing enough to readers of “Electrochem” and be published by “Electrochem” with minor revisions.
1. The authors mentioned that effective selectivity and specificity of electrochemical analysis are affected by the redox behavior of sensor elements against electrolyte/analyte sample state, pH, and interferents. For samples with different pH or composition, how to make sure the biosensor could be used?
2. The second half of the second paragraph and the third paragraph on page 7 seem to have different fonts and sizes than the main text. Moreover, the caption of Table 3 seems to have the same issue.
3. I think the most obvious advantage of the electrochemical-based biosensor system is its portability, which can facilitate semi in-situ detection.
4. Is there any study that combines multiple biosensors at the same chip system to detect several toxicants at the same time?
Comments on the Quality of English LanguagePage 2 Line 4 Although urbanization and industrial development improved the quality of human life at the same time left us several toxicant impurities in the environment.
Page 2 Line 29 nal transduction even at ultralow sample concentration. Depending on the experimental
Page 11 Line 1 sensor element against electrolyte/analyte sample state, pH and interferents.
Reviewer 2 Report
Comments and Suggestions for Authors
The present review is well-conceived and organised. It reports different kinds of electrochemical biosensors for the detection of toxicant compounds.
In my opinion, since these are electrochemical devices, it would be interesting for the authors also to address the aspect of interfering compounds present in the matrices described and what strategies were used to overcome this problem.
Reviewer 3 Report
Comments and Suggestions for Authors
The submitted review work “Electrochemical-based Biosensor Platform on Lab-Chip Model for Point-of-Need Toxicant Analysis “could be of relevant information and interest to the biosensors community that is working on the detection of toxicants. However, some work needs to be done in order to highlight and synthesize more of the relevant information that pretends to be provided to the scientific community with this review work. The comments listed below must be addressed before the manuscript can be accepted for publication. After addressing those major revisions, it can be considered for publication.
1. Page 3, check some typos of double time the same word “can be”.
2. In Table 1, what is the meaning of “hau”? Please indicate in the footnote.
3. Page 5, It is only mentioned: “Traditional methods for heavy metal detection in wastewater involve complex and time-consuming processes, often requiring sophisticated equipment and trained personnel”. Please provide examples and references of such traditional methods.
4. Page 6, “Li et al. developed a microbial biosensor comprised of E. coli immobilized with benzoquinone within a silica/gelatin hydrogel on the glassy carbon surface. Toxic metals such as Hg2+, Cd2+ and Cu2+ were detected from waste water using the above developed biosensor [47]. Electrochemical biosensor utilizing label-free method to monitor the exposure of heavy metals (Cr, Cd, Cu, Pb and Zn) in human cervical carcinoma (HeLa) cells using pencil graphite electrode [48]. Electrochemical DNA biosensor was also deployed for the detection of heavy metals, using the single stranded DNA as probe for selective binding of complementary target to form DNA duplex [49]”. Could you please shortly indicate what are the working detection principles of those biosensors?
5. Table 2, could you please add an extra column with the toxic dose?
6. Page 7, please check the size and style of the font since it is different from the previous pages.
7. Table 3, could you please add an extra column with the toxic dose?
8. Section 6., it is not clear what wants to be highlighted: the different electrochemical methods, the different platforms, the different receptors, the different ROS species? Please rework this section and state the purpose of the same in a very short sentence.
9. Page 10, what is the meaning of “barely”? maybe write a synonym.
Round 2
Reviewer 3 Report
Comments and Suggestions for Authors
The authors have addressed all the comments made. Therefore, I would suggest the publication of this manuscript.